# Study of the Impact of Industrial Restructuring on the Spatial and Temporal Evolution of Carbon Emission Intensity in Chinese Provinces—Analysis of Mediating Effects Based on Technological Innovation

**DOI:** 10.3390/ijerph192013401

**Published:** 2022-10-17

**Authors:** Jianshi Wang, Shangkun Yu, Mengcheng Li, Yu Cheng, Chengxin Wang

**Affiliations:** College of Geography and Environment, Shandong Normal University, Jinan 250358, China

**Keywords:** industrial structure rationalization, industrial structure optimization, technological innovation, carbon emission intensity, China

## Abstract

Global warming caused by greenhouse gas emissions seriously threatens a region’s sustainable environmental and socioeconomic development. Promoting industrial restructuring and strengthening technological innovation have become an important path to achieving pollution and carbon reduction as well as the green transformation of economic structure. This paper explored the mechanism of the mediating effect of technological innovation on industrial restructuring and carbon reduction while accounting for the direct effect of industrial restructuring on carbon emissions. Then, based on China’s provincial panel data from 2001 to 2019, we estimated the carbon emission intensity using the Intergovernmental Panel on Climate Change (IPCC)’s methods and analyzed its spatiotemporal evolution characteristics. Finally, we constructed a fixed-effect model and a mediating effect model to empirically analyze how industrial restructuring and technological innovation affect carbon emission intensity. The results are as follows: (1) From 2001 to 2019, China’s carbon emission intensity showed a continuous downward trend, with a pronounced convergence trend; there were obvious differences in carbon emission intensity between eastern, central, and western regions (western region > central region > eastern region) due to the unbalanced industrial structure. (2) In terms of direct effects, industrial restructuring can significantly reduce carbon emission intensity. The intensity of the effect is inversely proportional to the level of industrial restructuring, and the results of sub-regional tests are similar. Nevertheless, there is an obvious regional difference in the size of the carbon emission reduction effect of industrial restructuring in the east, central, and western regions. (3) In terms of indirect effects, industrial restructuring can reduce carbon emission intensity by enhancing technological innovation, and it acts as a mediating variable in the process of industrial restructuring to reduce carbon emission. Finally, we put forward recommendations for promoting industrial restructuring, strengthening green technological innovation, and properly formulating carbon reduction measures to provide a reference for countries and regions to achieve the goals of carbon neutrality, carbon peaking, and high-quality economic development.

## 1. Introduction

Since the industrial revolution, the world has witnessed rapid socioeconomic development, and the concentration of greenhouse gases such as carbon dioxide and the global average temperature have also risen sharply [1,2]. According to the “State of the Global Climate 2020” released by the World Meteorological Organization, the global average temperature in 2020 increased by about 1.2 °C compared to before the industrial revolution [3]. Climate warming has an irreversible multi-dimensional impact on the earth’s ecosystem and human habitat. The frequency and intensity of extreme weather events such as sea level rise, heat waves, droughts, and floods continue to increase. Together, these factors lead to potential conflict, friction, and even war between major global powers [4,5]. China’s carbon emissions have remained a challenging issue [6]. According to the relevant statistics provided in the BP Statistical Review of World Energy 2020, China’s total carbon emissions in 2019 ranked first in the world, accounting for 28.76% of the world’s total, more than the sum of the EU and the United States’ carbon emissions (24.28%) [7]. Facing the huge pressure of carbon reduction, China took the initiative to undertake international responsibilities in line with its national conditions and made a series of commitments to the international community. At the Paris Climate Conference in 2015, China proposed to achieve a carbon peak around 2030 and reduce the carbon emissions per unit GDP by 60%–65% compared with 2005 [8]. In 2020, President Xi Jinping promised in the general debate of the 75th United Nations General Assembly that China would strive to reach the peak of carbon emissions by 2030 and achieve carbon neutrality by 2060 [9].

Since the reform and opening up, China’s economic development has relied on an energy structure dominated by coal, and the industrial structure has become more industrialized, with high carbon emissions [10,11]. According to some scholars, the contribution rate of China’s industrial restructuring to the achievement of carbon intensity goals is more than 70% [12]. Industrial restructuring, together with more environmentally friendly and efficient energy-saving and emission-reduction technologies, can bring about advanced and new production technologies. The improvement of production technology and the increased technical content of products can effectively improve production efficiency, elevate energy efficiency, reduce factor inputs and pollution emissions per unit of output, and weaken the negative environmental externalities caused by production processes [13,14]. Therefore, industrial restructuring is one of the important means to achieve economic transformation and reduce carbon emissions [15]. China’s scientific and technological innovation capabilities have improved in recent years, making it the first driver for high-quality economic development. Technological innovation can reconfigure production factors, change the existing design, production processes, and technological processes of enterprises, develop and utilize low-carbon clean technologies, promote clean energy and new energy, improve the recycling efficiency of existing resources and energy, and reduce the emission intensity of unit carbon dioxide [16,17]. In view of this, this study takes 30 provinces in China as the research object (except Hong Kong, Macau, Taiwan and Tibet) and analyzes the relevant literature and theories in order to achieve the following objectives: this study takes carbon emission intensity as the explanatory variable, industrial structural transformation in each province as the core explanatory variable, and science and technology innovation as the mediating variable, and selects economic development, population size, urbanization rate and environmental regulation as control variables to clarify the direct and indirect effects of industrial structural transformation on carbon emission intensity, explore whether industrial structural transformation can achieve energy saving and emission reduction through science and technology innovation, and analyze the mediating role played by science and technology innovation in the model. 

The remainder of this article is organized as follows: Section 2 reviews the relevant literature; Section 3 presents the theoretical analysis and research hypotheses; Section 4 describes the data sources and evaluation models used for the present study; Section 5 exhibits the spatiotemporal evolution characteristics of China’s carbon emission intensity; Section 6 analyzes the research results; and Section 7 presents the conclusions and recommendations.

## 2. Literature Review

With the increasingly severe greenhouse gas effect, carbon emissions have drawn widespread attention from governments and scholars worldwide. Promoting industrial restructuring and improving technological innovation are considered the main measures to achieve energy conservation, emission reduction, and transformation of economic development models. At present, scholarly research on this aspect mainly focuses on the following two aspects:

The impact of industrial restructuring on carbon emissions has been analyzed mainly from the following three aspects. (1) The relationship between industrial restructuring and carbon emissions: most researchers believe that industrial restructuring can significantly reduce carbon emission intensity and optimize the regional environment [18,19], while some scholars hold that there is a significant nonlinear relationship between industrial structure and carbon emission intensity [20,21]. (2) The methods used to explore the impact of industrial restructuring on carbon emissions: the widely used method is the LMDI decomposition model. For example, Simbi et al. used this method to examine the main sources and potential driving forces of carbon dioxide in 20 African countries from 1984 to 2014. They found that the industrial structure and emission efficiency partially offset carbon dioxide growth [22]. The second approach uses a panel data model, spatial econometric model, and other measurement methods to empirically analyze the impact of industrial restructuring on carbon emissions [23,24]. For example, Cheng et al. utilized a dynamic spatial panel model to show spatial differences in the impact of industrial structure on China’s carbon intensity [25]. Other researchers have looked at the carbon reduction effect of industrial restructuring using the input–output table, STIRPAT, and other models [26,27,28]. (3) The impact of industrial restructuring on carbon emissions at different scales: Scholars have addressed this issue from the global, national, regional, and urban scales [29,30,31,32].

The research on the impact of technological innovation on carbon emissions can be reviewed from the following three aspects. (1) The carbon reduction effects of different technological innovations [33,34]: for example, You et al. investigated the direct and indirect effects of technological progress types on industrial carbon efficiency based on industrial data from 30 provinces in China from 2001 to 2016. From the perspective of direct effects, energy technology progress is more conducive to improving carbon efficiency than carbon emission technology progress, and neutral technological progress is more beneficial to carbon efficiency than capital-implied technological progress. In terms of indirect effects, the progress of capital technology affects the improvement of carbon efficiency. The progress of energy technology has a positive and significant impact on carbon efficiency through the green upgrade of the industrial structure [35]. (2) The impact of technological innovation on carbon emissions in different industries [36,37]: for example, Erdoğan et al. believe that long-term innovation has no significant impact on carbon emissions in energy, transportation, and other industries; the improvement of industrial innovation can reduce carbon emissions, while construction innovation will increase carbon emissions [38]. (3) The impact mechanism of technological innovation on carbon emissions at different scales [39,40,41]: for instance, Obobisa et al. used panel time series data from 2000 to 2018 to examine the impact of green technology innovation on CO_2_ emissions in 25 African countries. The estimated results showed that green technology innovation significantly reduced carbon emissions in these countries [42].

To sum up, scholars have conducted fruitful work both theoretically and practically on the impact and measurement of industrial restructuring and technological innovation on carbon emissions. However, most studies still focus on the relationship between industrial structure, technological innovation, and carbon emissions, and few have incorporated the three into one analytical framework. Due to the significant impact of industrial restructuring on technological innovation, the interaction path between the two has an important impact on reducing carbon emission intensity. Given this, the present study measured China’s carbon emission intensity, analyzed its spatiotemporal evolution characteristics, and accounted for the direct and indirect impact of industrial restructuring and technological innovation on carbon emission intensity using provincial panel data from 2001 to 2019. This study makes contributions from the following two aspects. Firstly, it incorporates industrial restructuring, technological innovation, and carbon emission intensity into one analytical framework and illuminates their relationships from the perspectives of industrial structure rationalization and optimization, which is theoretically helpful to clarify the mechanisms of action among the three. Secondly, this study employs the fixed effect model and the mediating effect model to empirically investigate the direct impact of industrial restructuring on carbon emissions and the mediating role of technological innovation in industrial restructuring affecting carbon emission intensity.

## 3. Theoretical Analysis and Research Hypotheses

### 3.1. Direct Mechanism of Industrial Restructuring on Carbon Emission Intensity

Industrial restructuring refers to the process or trend in which the industrial structure develops from unreasonable to rational and from low-level to high-level. It is the organic unity of industrial structure rationalization and industrial structure optimization. Therefore, this study analyzed the impact of industrial structure on carbon emission intensity from the perspectives of rationalization and optimization [11]. In terms of the impact of industrial structure rationalization on carbon emission intensity, as resources flow from sectors with lower productivity to sectors with higher productivity, low-end enterprises with extensive production methods and serious ecological damage in the industrial structure are eliminated, and they gradually evolve to eco-friendly, high-tech industrial sectors [16,27]. By adjusting the ratio between industries, the proportion of resource-intensive industries, labor-intensive industries, and other high-energy-consuming industries tends to decrease; the shares of technology- and knowledge-intensive industries are increased; resources within the industries are rationally allocated and effectively utilized, thereby promoting the production efficiency and resource utilization efficiency of enterprises and reducing the carbon emission intensity per unit product [25,43]. Regarding the impact of industrial structure optimization on carbon emission intensity, since most energy-intensive sectors are concentrated within the secondary industry, its carbon emissions are also the highest among the three industries [24]. As the industrial structure shifts towards low carbonization and service orientation, the output value of the tertiary industry continues to increase; the output value of the secondary industry gradually declines; the energy utilization efficiency improves; the consumption and demand for energy resources due to economic growth decrease; and the carbon emission intensity declines [17]. On the other hand, within the industrial structure, traditional industries and low-tech industries have enormous energy demands and significant carbon emissions due to the low level of technological innovation. Industrial structure optimization promotes the internal upgrading of the industrial structure whereby emerging and high-tech industries gradually replace traditional and low-tech industries. Production efficiency and energy utilization efficiency are correspondingly improved [14,18]. Therefore, without considering other goals, carbon emissions can be effectively reduced by restricting high-emitting industries and expanding low-emitting industries [44].

Therefore, we propose Hypothesis 1: Industrial structure rationalization and optimization have a significantly inhibitory effect on carbon emission intensity.

### 3.2. Indirect Mechanism of Industrial Restructuring on Carbon Emission Intensity

Industrial restructuring not only has a direct impact on carbon emission intensity but also indirectly affects it through technological innovation. From the perspective of factor allocation, the transformation and upgrading of industrial structure promote the reorganization and reallocation of production factors among different industries and sectors. Under the guidance of current policies and market mechanisms, production factors gradually shift from low-productivity sectors to high-productivity sectors [45]. In this process, the spatial reset and circulation of innovation resources will inevitably affect the efficiency of technological innovation. Sectors with high productivity often enjoy higher technical levels. When production factors gradually flow to these sectors, along with the circulation and concentration of innovation resources, the technical level between industries is further improved, thus driving the rapid development of high-tech industries. Industrial input has changed from high-carbon input to low-carbon input, and improving industrial output efficiency leads to an increase in unit energy output benefits, thereby reducing carbon emissions [34]. From the viewpoint of industrial linkages, the transformation and upgrading of industrial structure enhance technological innovation within and between industries through the accumulation of capital, labor, and technology flows [33]. Meanwhile, industrial restructuring has a remarkable radiation effect on the upstream and downstream industries through the correlation effect, improving the technological innovation capabilities of the upstream and downstream industries, resulting in the optimization of the industrial input–output structure of enterprises, and substantially reducing the factor inputs and carbon emissions per unit output [46]. From the standpoint of supply and demand, industrial restructuring promotes the upgrading of residents’ consumption demand, which increases the market demand for products with good environmental quality and high technology content [47,48]. Driven by the market and profits, enterprises will increase the intensity of R&D investment, enhance technological innovation, upgrade existing production equipment and production processes, introduce new, more advanced and environmentally friendly technologies and methods, and launch new processes and new products to meet the diversified and personalized product demand of the market. This will promote the transformation of enterprises from the original factor-driven model to the innovation-driven development model and indirectly promote carbon reduction [45,49].

Therefore, we propose Hypothesis 2: Industrial restructuring promotes technological innovation and acts on carbon emission intensity.

In summary, the influence path of industrial restructuring and technological innovation on carbon emissions can be summarized as the positive transmission mechanism of “industrial restructuring → technological innovation → carbon emission intensity reduction”, and the research logic model is shown in Figure 1.

## 4. Research Design

### 4.1. Modeling

In this study, the fixed-effect panel model was used to test the direct impact mechanism of industrial restructuring on carbon emission intensity. The benchmark regression model for the test is as follows:(1)$$CI_{it}=\alpha_{0}+\alpha_{1}INS_{it}+\alpha_{2}X_{it}+\mu_{i}+\varepsilon_{it}$$

In Equation (1), *CI_it_* represents the carbon emission intensity of province *i* in year *t*; *INS_it_* stands for industrial restructuring, which is decomposed into industrial structure rationalization (*RIS*) and industrial structure optimization (*OIS*); α_0_ refers to the intercept; *α*_1_ and *α*_2_ are the regression coefficients of *INS* and *X*, respectively; *μ_i_* expresses the individual effect; *ε_it_* is the random disturbance term; *X* means the relevant control variable.

As mentioned above, industrial restructuring not only directly impacts carbon emission intensity but may also indirectly impact the reduction of carbon emission intensity through technological innovation. To confirm whether technological innovation plays a mediating role in this process, the following mediating effect model was constructed about the study by Wen et al. [50].
(2)$$TEC_{it}=\beta_{0}+\beta_{1}INS_{it}+\beta_{2}X_{it}+\mu_{i}+\varepsilon_{it}$$
(3)$$CI_{it}=\lambda_{0}+\lambda_{1}INS_{it}+\lambda_{2}TEC_{it}+\lambda_{3}X_{it}+\mu_{i}+\varepsilon_{it}$$

In Equations (2) and (3), *TEC_it_* represents the technological innovation in year *t* of province *i*. Equation (2) is dedicated to the impact of industrial restructuring on technological innovation, and Equation (3) focuses on the combined impact of industrial restructuring and technological innovation on carbon emission intensity. If the coefficients *β*_1_ and *λ*_2_ in Equations (2) and (3) are significant at the same time, technological innovation is a mediating variable; if *λ*_1_ is not significant, technological innovation is a complete mediating variable; if *λ*_1_ is significant and is smaller than *λ*_3_, technological innovation is a partial mediating variable. The difference between the partial mediating effect and the complete mediating effect is that the core explanatory variable of the former can directly affect the explained variable. At the same, the latter must rely on the action of the mediating variable.

### 4.2. Variables

#### 4.2.1. Explained Variables

Carbon emission intensity (*CI*): Carbon intensity refers to carbon emissions per unit GDP. To date, there has been sufficient research on carbon emissions, but carbon intensity can reflect the actual economic development and CO_2_ emissions of a country (region) more scientifically than carbon emissions [51]. There is no unified measurement method for carbon emissions. Therefore, the annual carbon emissions of each province (Table 1) were estimated based on the carbon emission coefficients of eight types of fossil fuels provided in the “2006 IPCC Guidelines for National Greenhouse Gas Inventories” and “China Energy Statistical Yearbook” as well as the energy consumption of fossil fuels in each province using Equation (4). On this basis, the carbon emissions of each province (city and autonomous region) were obtained.
(4)$$C_{co_{2}}=k\cdot \sum_{i=1}^{n}E_{i}\cdot \delta_{i}$$

In Equation (4), *C_co2_* is the carbon dioxide emissions; *k* (*k* = 44/12) refers to the carbon dioxide to carbon molecular weight ratio; *E_i_* represents the consumption of the *i^th^* fossil fuel; *δ_i_* stands for the emission coefficient of the *i^th^* fossil fuel.

#### 4.2.2. Explanatory Variables

Industrial restructuring: Industrial restructuring was decomposed into two parts: industrial structure optimization (*OIS*) and industrial structure rationalization (*RIS*) [52]. RIS is expressed by the Theil index using the following Equation:(5)RIS=∑i=1n(YiY)ln(YiLiYL)=∑i=1n(YiY)ln(YiYiLiL)

In Equation (5), *Y* is the industrial output value; *L* represents the number of employees in the industry; *i* stands for the industry; *n* expresses the number of industrial sectors; *Y/L* stands for the productivity; *Y_i_/Y* means the output structure; *L_i_/L* refers to the employment structure. According to the basic assumptions of classical economics, when the economy is in equilibrium, the efficiency of industrial sectors is at an equalized level, *Y_i_/L_i_* = *Y/L*, namely, *RIS* = 0. Conversely, if *RIS* is not equal to 0, the more the industrial structure deviates from the equilibrium, the more unreasonable the industrial structure is. Since this indicator is negative, it was reciprocated in the subsequent regression. The ratio of the added value of the tertiary industry and that of the secondary industry in a certain region was used as a proxy indicator for *OIS*. If the value is rising, the regional economy is moving towards servitization, and the industrial structure is more advanced.

#### 4.2.3. Mediating Variables

Technological innovation (*TEC*): Technological innovation refers to the invention and creation ability of a region, enterprise, or individual in a certain technical field [33]. The number of patents can indicate the level of invention and innovation and the degree of innovation activity of a region or enterprise and reflect the real technological innovation achievements of a specific region in that year. The data are accurate, authoritative, and easy to obtain. In this study, the number of patent applications accepted (take the logarithm) was used to measure the level of technological innovation.

#### 4.2.4. Control Variables

Previous studies have established that a country or region’s carbon emission intensity is affected not only by industrial restructuring and technological innovation but also by population size, the urbanization process, economic development level, the intensity of environmental regulation, and other factors [53,54]. In this study, population size (*POP*), urbanization rate (*URB*), economic development (*RGDP*), and environmental regulation (*ER*) were used as control variables that affect the intensity of regional carbon dioxide emissions. 

The population affects urban infrastructure construction and industrial development through the agglomeration effect and scale effect, which in turn affect the carbon emission intensity effect. The number of permanent residents represents the population at the end of the year. 

Urbanization leads to the improvement of urban infrastructure and environmental management. On the other hand, it also leads to numerous urban problems. Urbanization is expressed by the proportion of the permanent urban population to the total population.

According to the environmental Kuznets hypothesis, the level of regional economic development is closely related to environmental pollution. The environmental quality will show an inverted U-shaped trend of deterioration followed by improvement as the level of economic development increases. RGDP is represented by the real GDP per capita.

The implementation of environmental regulations can force enterprises to adopt cleaner production methods and more advanced energy-saving and emission-reducing technologies for production to reduce the cost of environmental management and the demand for high-carbon energy. Environmental regulation is expressed by the proportion of industrial pollution control investment in GDP. 

### 4.3. Data Sources and Descriptive Statistics

In view of the differences in statistical calibers and methods of data from Hong Kong, Macao, and Taiwan, and the severe lack of data from Tibet, and considering the completeness and availability of data, this study finally selected 30 provinces (municipalities and autonomous regions) in China as the study area. The data were derived mainly from “China Statistical Yearbook”, “China Industrial Statistical Yearbook”, “China Environmental Statistical Yearbook”, “China Science and Technology Statistical Yearbook”, “China Energy Statistical Yearbook”, the website of the National Bureau of Statistics, and the statistical yearbooks of these provinces (municipalities and autonomous regions) between 2002 and 2020. For the missing data of individual years, the interpolation method was used to fill in the data, and the data on price variables were deflated with 2001 as the base period. According to research needs, the 30 provinces (cities and autonomous regions) were divided into the eastern region (Beijing, Tianjin, Hebei, Liaoning, Shanghai, Jiangsu, Zhejiang, Fujian, Shandong, Guangdong, Hainan), central region (Jilin, Heilongjiang, Shanxi, Anhui, Jiangxi, Henan, Hubei, Hunan), and western region (Inner Mongolia, Guangxi, Chongqing, Sichuan, Guizhou, Yunnan, Shaanxi, Gansu, Qinghai, Ningxia, Xinjiang). Table 2 shows the descriptive statistics of each variable.

## 5. Spatiotemporal Evolution Characteristics of Carbon Emission Intensity

From the perspective of temporal evolution, the carbon emission intensity of the whole country and the three major regions showed a continuous downward trend with certain stages from 2001 to 2019 (Figure 2). The national carbon emission intensity dropped from 4.50 t/CNY 10,000 to 2.68 t/CNY 10,000, with an average annual decrease of 2.84%. To be specific, it dropped from 3.32 t/CNY 10,000 to 1.76 t/CNY 10,000, with an average annual decrease of 3.48% in the eastern region; it dropped from 5.20 t/CNY 10,000 to 2.35 t/CNY 10,000, with an average annual decrease of 4.32% in the central region; it dropped from 5.18 t/CNY 10,000 to 3.86 t/CNY 10,000, with an average annual decrease of 1.63% in the western region. In terms of stages, 2001–2006 was a relatively stable stage, and the decline in carbon emission intensity in the whole country and the three major regions was the same with little change; 2006–2011 was a slow decline stage, with an average annual decline of 3.78% in the national carbon emission intensity, specifically, 3.54% in the eastern region, 5.78% in the central region and 2.83% in the western region; 2011–2019 was a stage of rapid decline, with an average annual decline of 4.66% in the national carbon emission intensity, specifically, 5.34% in the eastern region, 5.37% in the central region and 3.99% in the western region.

The overall carbon emission intensity in the eastern region was relatively low and always lower than the national average level. The carbon emission per unit in the central region was relatively high before 2008. After 2008, the average annual decrease in carbon emission per unit was faster and lower than the national average level. The carbon emission intensity in the western region was relatively high and always higher than the national average level. The overall carbon emission intensity of the three major regions presented a pattern of “western region > central region > eastern region”, with regional non-equilibrium characteristics, but the regional differences in carbon emission intensity were gradually narrowing, and the convergence trend was becoming more and more obvious. On the whole, China’s carbon emission intensity continued to decrease as a whole from 2001 to 2019, indicating that China’s green economic transformation, energy conservation, and emission reduction policies were productive, laying a solid foundation for high-quality economic development in the new era. In particular, since the new development concepts of “innovation, coordination, green, openness, and sharing” were put forward at the 18th National Congress of the Communist Party of China, governments at all levels have vigorously promoted industrial restructuring, eliminated outdated process equipment, promoted energy-saving technology products, and built clean, low-carbon, safe and efficient energy systems, thereby improving the efficiency of energy utilization and reducing the carbon emission intensity per unit of GDP continuously.

To explore the spatial evolution characteristics of China’s carbon emission intensity, a K-means cluster analysis was carried out using SPSS26.0 based on the level of carbon emission intensity in each province from 2001 to 2019. Then, ArcGIS10.8 was employed to visualize the carbon emission intensity of the 30 provinces. The carbon emission intensity was divided into four types according to the clustering results: low-level area, relative low-level area, relative high-level area, and high-level area (Figure 3). From 2001 to 2019, China’s carbon emission intensity showed a spatial pattern of high in the west and low in the east, high in the north and low in the south. The provinces with high carbon emission intensity were mainly concentrated in the northern and northwestern regions, while the carbon emission intensity in the eastern region was relatively low. From the perspective of different years, the regional differences in carbon emission intensity in 2001 were obvious, and the spatial agglomeration characteristics were significant, mainly expressed in low-level and relative high-level areas. There were 13 provinces in the low-level area, mainly located in the coastal areas and the Sichuan-Chongqing area. There were ten provinces with relatively high carbon emission intensity, mainly in the central and northeastern regions. Shanxi Province was a low-level area, while Guizhou, Ningxia, Liaoning, Gansu, Inner Mongolia, and Xinjiang were relatively high-level. The change in the overall spatial pattern in 2007 was not obvious compared with 2001, and low-level areas and relative high-level areas still dominated it. Shandong, Yunnan, and Hainan changed from low-level areas of carbon emission intensity to high-level areas, while Shanghai changed from a relatively high-level area to a high-level one. The spatial differentiation pattern of carbon emission intensity changed significantly in 2013 compared with 2007. The low-level areas increased significantly from 11 to 20, whereas the high-level areas decreased remarkably from 12 to 6 and gradually shifted and shrank to the western inland areas. In 2019, the low-level carbon areas further increased to 23 from 20 in 2013. The number of low-level areas did not change. The high-level areas mainly included Liaoning, Inner Mongolia, Shaanxi, Gansu, and other provinces. Shanxi and Xinjiang had relatively high carbon emission intensity and were considered high-level areas.

## 6. Empirical Results

### 6.1. Benchmark Regression Results

To represent the correlation between industrial restructuring and carbon emission intensity more vividly, a scatter-fitting diagram of industrial structure rationalization, industrial structure optimization, and carbon emission intensity was drawn (Figure 4). According to the trend of the fitting line, there was a negative correlation between industrial structure rationalization, industrial structure optimization, and carbon emission intensity, which could support Hypothesis 1 to a certain extent. However, the real impact of the two on carbon emission intensity still needs to be further tested using a model.

The present study used a panel model to verify the impact mechanism of industrial restructuring on carbon emission intensity. LM and Hausman tests revealed that the fixed-effect model was better than the mixed effect and random effect models, and the fixed-effect model was the optimal explanatory model. To ensure the stability of the panel regression results, the estimated results of random effects and OLS were also listed (Table 3).

Judging from the estimation results at the national level, industrial structure rationalization had a significant negative impact on carbon emission intensity. With other conditions remaining unchanged, every 1% increase in the level of industrial structure rationalization reduced carbon emission intensity by 0.547%. This showed that industrial structure rationalization promoted the benign flow of production factors among various industrial sectors, improved the effective allocation and rational utilization of resources, enhanced the coupling quality of the input structure and output structure of inter-industry factors, and significantly elevated the efficiency of resource utilization and the level of comprehensive utilization, thereby effectively reducing carbon emission intensity. Industrial structure optimization had a significant negative impact on carbon emission intensity. When related variables were controlled, every 1% increase in the level of industrial structure optimization reduced the carbon emission intensity by 1.117%. This indicated that the optimization of the industrial structure led to the gradual elimination or transfer of industries with high pollution and high energy consumption and the accelerated development of high value-added and low energy consumption industries such as producer services and modern manufacturing, thus improving the energy structure, reducing fossil energy consumption, and lowering carbon emission intensity. Therefore, industrial structure rationalization and industrial structure optimization could dramatically reduce the carbon emission intensity, but from the panel estimation results, industrial structure optimization had better carbon reduction effects than industrial structure rationalization. For this reason, in the process of regional industrial restructuring, it is necessary not only to improve the utilization efficiency and coupling degree of inter-industry factor resources and promote the coordinated and balanced development of various industries and within industries but also to accelerate the development of modern service industries, high-tech industries, and strategic emerging industries and improve the advanced level of the industrial structure.

From the perspective of control variables, the driving effect of population size on carbon emission intensity was very significant, passing the 1% significance test, indicating that with the expansion of population size, the energy demand for production and living increased, thus increasing the unit carbon emissions. The impact of urbanization rate on carbon emission intensity was significantly negative, showing that with the increase in urbanization rate, the agglomeration effect and scale effect of urbanization gradually appeared, which showed a significant inhibitory effect on carbon emission intensity. Economic development had an obvious negative inhibitory effect on carbon emission intensity, indicating that the improvement in economic development level drew more funds to be invested in the research and development of energy conservation, emission reduction, and cleaner production, effectively reducing carbon emission intensity. However, the effect of environmental regulation on carbon emission intensity did not pass the significance test, exhibiting that current environmental regulation was weak and the effect on carbon emission had not yet appeared.

### 6.2. The Mediating Effect of Technological Innovation

The mediation effect test was further used to analyze the transmission mechanism of the impact of industrial restructuring on carbon emission intensity. The panel fixed effect model was employed to explore the mediating effect of technological innovation and verify whether it acts as a mediating variable in industrial restructuring affecting carbon emission intensity. The first step was the regression result of Equation (1), see Table 3, and the second and third steps were the regression of Equations (2) and (3), see Table 4. Columns 1 and 2 are the regression results of industrial structure rationalization and industrial structure optimization on technological innovation, and columns 3 and 4 are the regression results of industrial structure rationalization and technological innovation, industrial structure optimization, and technological innovation on carbon emission intensity. 

Judging from the regression results of columns 1 and 2 in Table 4, the impact of industrial structure rationalization and industrial structure optimization on technological innovation passed the 1% significance test. The influence coefficients were 0.244 and 0.387, respectively. In other words, every 1% increase in industrial structure rationalization and industrial structure optimization increased technological innovation by 0.244% and 0.387%, respectively, indicating that the core explanatory variables had a significant impact on the explained variables; that is, both industrial structure rationalization and industrial structure optimization could significantly promote regional technological innovation. Judging from the regression results in columns 3 and 4, the regression coefficients of technological innovation passed the 1% and 5% significance tests, and the influence coefficients were –0.527 and –0.273, respectively, showing that technological innovation had a significant inhibitory effect on carbon emission intensity. The regression coefficients of industrial structure rationalization and industrial structure optimization were –0.429 and –1.011, and both passed the 1% significance test, indicating that technological innovation was mediating in the process of industrial restructuring affecting carbon emission intensity. Industrial restructuring could improve the energy structure and reduce energy consumption by promoting technological innovation, thereby significantly inhibiting the unit carbon emission intensity. As such, Hypothesis 2 proves valid.

### 6.3. Regional Differences

Since there are significant differences in industrial resource endowments, industrial bases, and economic development levels in different regions, the present study further explored the impact of industrial restructuring on carbon emission intensity in the three major regions. Judging from the estimation results at the regional level (Table 5), at the 1% significance level, the estimation results of both industrial structure rationalization and industrial structure optimization passed the significance test. The influence coefficients of industrial structure rationalization on the carbon emission intensity of the three regions were negative, at −0.169, −3.566, and −2.400, respectively, indicating that for every 1% increase in industrial structure rationalization, the carbon emission intensity of the three regions decreased by 0.169%, 3.566%, and 2.400%, respectively. The influence coefficients of industrial structure optimization on the carbon emission intensity of the three major regions were negative, which were −1.939, −2.197, and −1.815, respectively, showing that for every 1% increase in industrial structure optimization, the carbon emission intensity in the three regions dropped by 1.939%, 2.197%, and 1.815%, respectively.

It could also be seen from the table that industrial structure rationalization and industrial structure optimization had a greater impact on carbon emission intensity in the central and western regions than in the eastern region. On the one hand, industrial restructuring had a marginal decreasing effect on carbon emission intensity, which weakened with the continuous optimization of the industrial structure. Compared with the eastern region, the industrial level in the central and western regions was relatively low; the industrial structure was relatively simple; the secondary industry had dominated for a long time; the secondary industry had more high-carbon sectors and was thus more likely to be positively affected by the energy-saving and emission-reduction effects brought about by changes in the industrial structure. Therefore, the impact of industrial restructuring on the carbon emission intensity of the central and western regions was more significant. On the other hand, most provinces in the eastern region had low carbon emission intensity, with little room for further decline, and the carbon reduction effect of industrial restructuring had been released in advance. In contrast, the economic foundation of the central and western regions was relatively weak; industry was still in the early stage of transformation; economic development was still highly dependent on traditional energy and high-energy-consuming industries; and there was still much room for the reduction of carbon emission intensity.

### 6.4. Robustness Test

The econometric model constructed herein may have endogeneity problems caused by variable omission, measurement error, and bidirectional causality, which might affect the robustness of the regression results. Therefore, to ensure the reliability of the regression results, the one-period lag of each explanatory variable was used as an instrumental variable, and the two-stage least squares method was utilized to test the robustness of the overall regression results and the mediating effect regression results (Table 6). From the regression results of robustness, it could be seen that the direction and significance of the regression coefficients of the benchmark regression test (model 1–model 2) and the mediating effect test (model 3–model 4) were not significantly different from the previous ones, indicating that the empirical conclusions drawn above were robust and reliable.

## 7. Discussions

From the direct effect, the relationship between industrial restructuring and carbon emission intensity is negative, which implies that industrial restructuring plays a positive role in the improvement of carbon emission reduction. This conclusion is supported by the results of Dong al. and Zhang [52,55]. From the general rule of industrial structure evolution, the shift of industrial structure center of gravity from primary to tertiary industries is often accompanied by the gradual replacement of traditional industries by low-pollution, low-energy-consuming and high-value-added new industries, which not only improves factor production efficiency but also reduces energy consumption intensity, thus positively influencing the reduction of carbon emissions [13,14,20]. Therefore, it is recommended that Chinese provinces continue to accelerate the pace of industrial transformation and upgrading, promote the industrial structure in the direction of advanced development, encourage the transformation and upgrading of traditional industries, improve industrial efficiency, and strive to achieve low energy consumption and high output while accelerating the formation of knowledge- and technology-intensive industries such as strategic emerging industries and high-tech industries.

In terms of the mediating effect, the test results of this study confirm that industrial restructuring has an indirect effect on carbon emission intensity through technological innovation. Industrial restructuring can re-match innovation resources, change the factor supply and demand structure, and force enterprises to improve science and technology innovation and the transformation ability of science and technology achievements by increasing R&D investment [17,56]. Technological innovation will promote substituting other factors such as new energy and capital for traditional energy through the transformation of results, which will reduce energy consumption [25,35]. Therefore, we suggest paying attention to the interaction between technological innovation and industrial restructuring to reduce carbon emission intensity jointly. At the same time, since there are obvious differences in the regional development of the eastern, central and western regions, it is necessary to grasp their development positioning accurately, combine the actual economic and social development of the region, and formulate a targeted carbon emission intensity control strategy in line with the region. The eastern region should fully play the core role of technological innovation in industrial structure upgrading while pursuing advanced industrial structure. The central and western regions should accelerate the enhancement of enterprise innovation capacity, focus on introducing production capital and technology-intensive industries, and use technology to drive the transformation and upgrading of industrial structure.

Of course, there are certain limitations to this paper. First, low-carbon pilot industrial restructuring may also affect carbon emissions through other indirect channels, and this paper only chose technological innovation in its examination. Industrial restructuring may also affect carbon emissions through channels such as government preferences and fiscal decentralization. Second, although this paper applied econometric models as much as possible and considered a series of factors that may influence the estimation results, it still could not fully solve the endogeneity problem of the model. Furthermore, due to the difficulty of data collection, Tibet, Hong Kong, Macau, and Taiwan were not included in this paper. Future research on the effect of industrial restructuring and technological innovation on carbon emissions needs to be followed up continuously. Finally, further empirical investigation on regional and administrative hierarchy heterogeneity is still needed due to the limitation of space. 

## 8. Conclusions

This study analyzed the spatiotemporal evolution characteristics of carbon emission intensity in 30 Chinese provinces using ArcGIS and the energy consumption data from 2001 to 2019, concerning the IPCC carbon emission accounting method. Industrial restructuring, technological innovation, and carbon emission intensity were incorporated into a unified research framework. The panel fixed effect and mediating effect models were used to empirically test the direct and indirect driving mechanisms of industrial restructuring and technological innovation on carbon emission intensity. The main conclusions are as follows. First, from 2001 to 2019, China’s global carbon emission intensity showed a continuous decline. Second, the results of the benchmark regression showed that the rationalization of industrial structure and the advanced industrial structure have significant adverse inhibitory effects in influencing carbon emission intensity, and there is regional heterogeneity. Third, The mediating effect analysis revealed that technological innovation was an important channel for industrial restructuring to affect carbon emission intensity. Industrial restructuring reduced carbon emission intensity through direct effects and the indirect effect of technological innovation.

## Figures and Tables

**Figure 1 ijerph-19-13401-f001:**
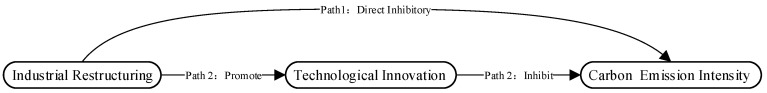
The research logic model.

**Figure 2 ijerph-19-13401-f002:**
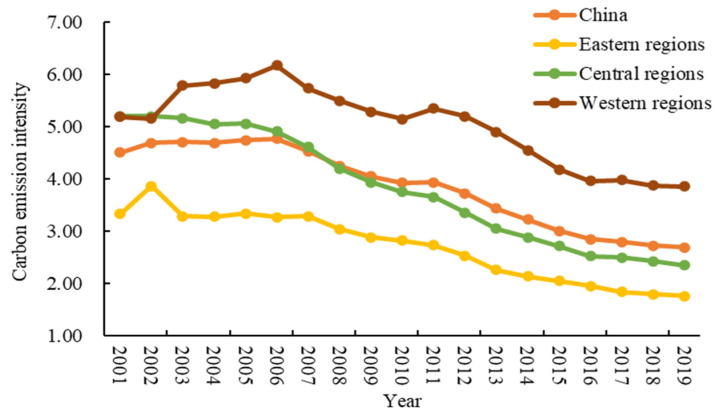
Variation trend of carbon emission intensity in China and the three major regions (2001–2019).

**Figure 3 ijerph-19-13401-f003:**
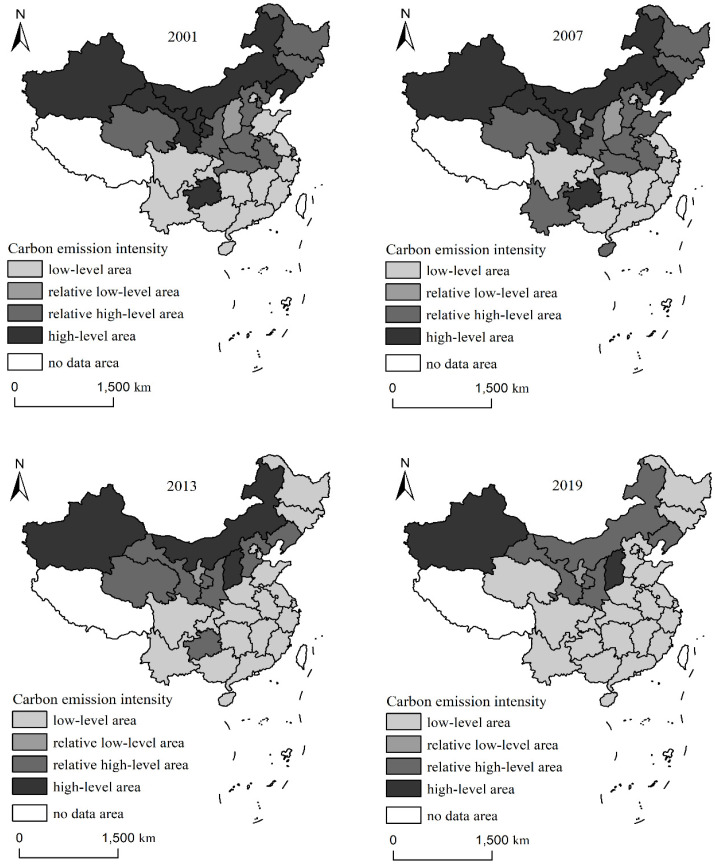
The spatial distribution pattern of carbon emission intensity in China (2001–2019).

**Figure 4 ijerph-19-13401-f004:**
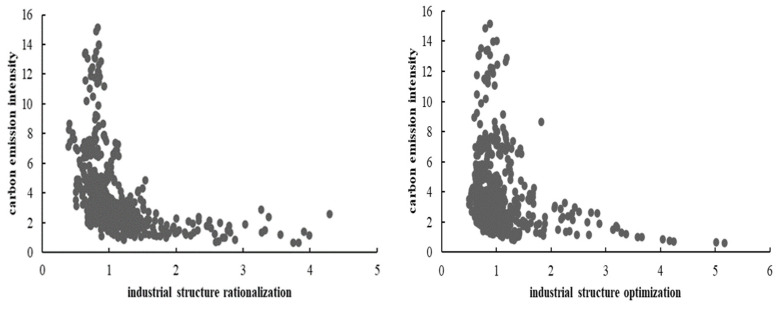
Linear fitting of industrial structure rationalization, industrial structure optimization, and carbon emission intensity.

**Table 1 ijerph-19-13401-t001:** The carbon emission coefficients of various fossil fuels.

Fuel Type	Carbon Content (kgc/GJ)	Carbon Oxidation Rate	Average Low Calorific Value (KJ/kg, m^3^)	Carbon Emission Coefficient (kgc/kg, m^3^)
Coal	25.8	1	20 908	0.539
Coke	29.2	1	28 435	0.830
Crude oil	20.0	1	41 816	0.836
Gasoline	18.9	1	43 070	0.814
Kerosene	19.6	1	43 070	0.844
Diesel	20.2	1	42 652	0.861
Fuel oil	21.2	1	41 816	0.882
Natural gas	15.3	1	38 931	0.595

Note: The data are from the “2006 IPCC Guidelines for National Greenhouse Gas Inventories” and “China Energy Statistical Yearbook”.

**Table 2 ijerph-19-13401-t002:** The descriptive statistics of variables.

Variable	Name	Sample Size	Mean	SD	Min	Max
Explained variable	CI	570	3.851	2.654	0.630	15.140
Explanatory variable	RIS	570	1.097	0.550	0.390	4.290
	OIS	570	1.060	0.563	0.500	5.170
Mediating variable	TEC	570	9.623	1.723	4.820	13.600
Control variable	POP	570	8.166	0.756	6.260	9.430
	URB	570	0.500	0.150	0.230	0.900
	RGDP	570	9.780	0.645	7.990	11.360
	ER	570	11.590	1.121	6.910	14.160

**Table 3 ijerph-19-13401-t003:** The overall regression results.

Variable	Fixed Effect	Random Effect	OLS	Fixed Effect	Random Effect	OLS
RIS	−0.547 ***	−0.436 ***	−0.436 **			
	(−3.53)	(−2.85)	(−2.85)			
				−1.117 ***	−0.873 ***	−0.870 ***
OIS				(−8.56)	(−6.89)	(−6.89)
POP	1.792 ***	0.318	−0.318	3.260 ***	0.184	0.184
	(2.77)	(0.80)	(−0.80)	(5.08)	(0.45)	(0.45)
URB	−7.110 ***	−5.534 ***	−5.530 ***	−6.537 ***	−4.654 ***	−4.653 ***
	(−4.55)	(−3.73)	(−3.73)	(−4.41)	(−3.23)	(−3.23)
RGDP	−0.411 *	−0.506 **	−0505 **	−0.487 **	−0.583 ***	−0.583 **
	(−1.77)	(−2.23)	(−2.23)	(−2.21)	(−2.65)	(−2.65)
ER	−0.009	−0.007	0.006	−0.001	−0.003	−0.003
	(−0.25)	(−0.20)	(0.20)	(−0.04)	(−0.10)	(−0.10)
Con	−2.713	14.561 ***	14.561 ***	−13.542 **	11.341 ***	11.340 ***
	(−0.50)	(4.23)	(4.23)	(−2.56)	(3.28)	(3.28)

Note: ***, **, and * indicate significance at the 1%, 5%, and 10% levels, respectively, here and below.

**Table 4 ijerph-19-13401-t004:** Regression results from the mediating effect of technological innovation.

Variable	TEC	TEC	CI	CI
RIS	0.224 ***		−0.429 ***	
	−4.19		(−2.76)	
OIS		0.387 ***		−1.011 ***
		−8.55		(−7.30)
TEC			−0.527 ***	−0.273 **
			(−4.27)	(−2.20)
Con	−14.090 ***		−10.137 *	−16.442 ***
	(−7.56)		(−1.81)	(−3.02)
Control variable	Yes	Yes	Yes	Yes
N	570	570	570	570

Note: ***, **, and * indicate significance at the 1%, 5%, and 10% levels, respectively, here and below.

**Table 5 ijerph-19-13401-t005:** Regression results by region.

Variable	Eastern		Central		Western	
RIS	−0.169 ***		−3.566 ***		−2.410 ***	
	(−1.19)		(−5.35)		(−4.44)	
OIS		−1.939 ***		−2.197 ***		−1.815 ***
		(−6.05)		(−6.80)		(−6.32)
Con	25.551 ***	−1.553	121.173 ***	125.449 ***	−78.858 ***	−72.032 ***
	(4.88)	(−0.19)	(4.16)	(4.82)	(−6.33)	(−5.99)
Control variable	Yes	Yes	Yes	Yes	Yes	Yes
N	209	209	152	152	209	209

Note: ***, **, and * indicate significance at the 1%, 5%, and 10% levels, respectively, here and below.

**Table 6 ijerph-19-13401-t006:** The robustness test results.

Variable	Model 1	Model 2	Model 3	Model 4
RIS	−0.961 ***		−0.971 ***	
	(−5.00)		(−5.33)	
OIS		−0.808 ***		−0.769 ***
		(−5.03)		(−4.82)
TEC			−0.706 ***	−0.656 ***
			(−3.51)	(−3.25)
Con	22.1254 ***	24.648 ***	8.415 *	11.809 **
	(10.29)	(11.82)	(1.83)	(2.57)
Control variable	Yes	Yes	Yes	Yes
N	570	570	570	570

Note: ***, **, and * indicate significance at the 1%, 5%, and 10% levels, respectively, here and below.

## Data Availability

The data presented in this study are available on request from the corresponding author.

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
