# Peer review of "Study of the Impact of Industrial Restructuring on the Spatial and Temporal Evolution of Carbon Emission Intensity in Chinese Provinces—Analysis of Mediating Effects Based on Technological Innovation"

_ijerph, 2022, doi:10.3390/ijerph192013401_

Round 1

Reviewer 1 Report

The manuscript explores the effect associated with technological innovation on the spatiotemporal evolution of carbon emission intensity. This manuscript has a panel dataset, almost 20 years of investigating intergovernmental panel on climate change. It also has an in-depth literature review which connects the drivers with carbon emission intensity: one by one. However, the manuscript has some bottlenecks settled as follows: 

1.    How is technological innovation measured? Is Oslo manual taken into account? Are these measures created at corporate level and then aggregated at province level?  If so, how is the aggregation procedure done? It would be also great to support the proxy for industrial structure optimization by economic reasons and references.

2.    It is still not clear how exactly industrial restructuring and technological innovation influence carbon emission intensity. In all the hypotheses, the authors could not link properly/explain clearly the link between the two variables being tested in each hypothesis. 

3.    Why are such control factors used for testing their effect on carbon emission intensity? What about the controls at industry, corporate, province level? Some economic reasoning should be provided.

4.    Why aren’t lagged explaining variables tested? Why do the authors assume that effect is synchronized?

5.    Why do authors exclude 4 Chinese provinces from research? 

6.    What model does better prediction? If the number of provinces doesn’t change, it seems that the panel data with fixed model should work better. Why do authors decide not to compare three models for panel dataset: fixed effect, random effect, pooled regression?

7.    Is it possible to check whether the model is robust to possible measurement biases? For example, R&D expenditures might be treated as proxy for technological innovation? Any time specific biases observed? 

8.    How can the Chinese experience be helpful to other countries for diminishing carbon emission? 

Author Response

Dear Editors and Reviewers:

Thank you very much for your comments about our paper submitted to International Journal of Environmental Research and Public Health (paper ID: ijerph-1921497).

We have learned much from your comments, which are fair, encouraging and constructive. After carefully studying the comments, we have made corresponding changes. Our response of the comments is enclosed at the end of this letter.

If you have any question about this paper, please don’t hesitate to contact us.

Sincerely yours.

The details of the revisions are as follows:

Point 1: . How is technological innovation measured? Is Oslo manual taken into account? Are these measures created at corporate level and then aggregated at province level?  If so, how is the aggregation procedure done? It would be also great to support the proxy for industrial structure optimization by economic reasons and references.

Response 1: This is a very helpful question for improving the logicality of our manuscript. Typically, scientific and technological innovation refers to the ability of a region, an organization, or an individual to invent and create in a particular discipline and technological area. The number of patents can provide an indication of the level of invention and creation and the degree of innovation activity in a region or organization. It is generally believed that the number of patents is an objective measure of the technological advancements of a region in that particular year, and this data is reliable, authoritative, convenient, and easy to obtain. According to previous studies, the number of patent applications accepted is a good indicator of the level of scientific and technological innovation, and the number of patent applications accepted is greater than the number of applications in China. As a result, the research does not use the Oslo manual to measure technological innovation.

Point 2: It is still not clear how exactly industrial restructuring and technological innovation influence carbon emission intensity. In all the hypotheses, the authors could not link properly/explain clearly the link between the two variables being tested in each hypothesis.

Response 2: Thank you very much for your suggestion, which has improved our article a lot. To clarify the relationship between industrial structure transformation and scientific and technological innovation on emission intensity, we add a schematic diagram. (Line 254-260, Page 6)

Point 3: Why are such control factors used for testing their effect on carbon emission intensity? What about the controls at industry, corporate, province level? Some economic reasoning should be provided.

Response 3: Thank you for your professional and constructive suggestion, it is very beneficial to enhance the credibility of our manuscript. As you suggested, in order to alleviate the problem of biased regression results due to omitted variables in the model, and with reference to relevant studies by scholars such as this one, economic factors, demographic factors, and governance factors are integrated into the research model as control variables in this paper. We add the expected effects of population size, urbanization rate, economic development, and environmental regulation on carbon emission intensity. (Line 345-362, Page 8)

Point 4: Why aren’t lagged explaining variables tested? Why do the authors assume that effect is synchronized?

Response 4: We are grateful for your detailed and professional review of our article. For the convenience of reading, from the input-output perspective, carbon emissions are generated in the process of production and life, thus carbon emissions itself can be regarded as an output. Output is based on inputs, and the development trend of industrial structure is the key element that affects and determines carbon emissions. Thus, the adjustment of industrial structure and the treatment of carbon emissions are synchronous and synergistic, so we assume that the effects are synchronous.

Point 5: Why do authors exclude 4 Chinese provinces from research?

Response 5: Thank you for your precious comments and advice. In view of the differences in statistical calibers and methods of data from Hong Kong, Macao, and Taiwan, and the serious lack of data in Tibet, and considering the completeness and availability of data, this study finally selected 30 provinces (munic-ipalities and autonomous regions) in China as the study area.

Point 6: What model does better prediction? If the number of provinces doesn’t change, it seems that the panel data with fixed model should work better. Why do authors decide not to compare three models for panel dataset: fixed effect, random effect, pooled regression?

Response 6: We are extremely grateful to the reviewer for pointing out this problem. The study uses a panel model to verify the mechanism of the effect of industrial structural transformation on carbon emission intensity, and the LM and Hausman tests find that the fixed-effects model outperforms the mixed-effects and random-effects models, and the fixed-effects is the optimal explanatory model. And, we have supplemented the results of OLS estimation of mixed effects estimation in Table 3 as you suggested. (Line 518-519, Page 13)

Point 7: Is it possible to check whether the model is robust to possible measurement biases? For example, R&D expenditures might be treated as proxy for technological innovation? Any time specific biases observed?

Response 7: We deeply appreciate the reviewer’s suggestion. In this paper, we mainly consider that the econometric model constructed herein may have endogeneity problems caused by variable omission, measurement error, and bidirectional causality, which might affect the robustness of the regression results. Therefore, to ensure the reliability of the regression results, the one-period lag of each explanatory variable was used as an instrumental variable, and the two-stage least squares method was utilized to test the robustness of the overall regression results and the mediating effect regression results (Table 6). From the regression results of robustness, it could be seen that the direction and significance of the regression coefficients of the benchmark regression test (model 1-model 2) and the mediating effect test (model 3-model 4) were not significantly different from the previous ones, indicating that the empirical conclusions drawn above were robust and reliable. (Line 595-605, Page 15-16)

Point 8: How can the Chinese experience be helpful to other countries for diminishing carbon emission?

Response 8: We are grateful for this detailed advice. In order to reduce carbon emissions, China has taken several measures. Climate change issues have been incorporated into national development plans, for example. As an alternative, the government has established important targets and promotional policies to reduce fossil fuel consumption and promote the use of low-carbon energy. Moreover, the Chinese government has made efforts to restructure the economy, develop an environmentally friendly economy, and maintain the flexibility of the economic system. As China develops, it contributes to the world by reducing greenhouse gas emissions and sharing its experience in this area with developing nations. In order to improve underdeveloped countries' ability to cope with climate change and adapt to it, international cooperation and knowledge exchange are essential. China has also initiated several actions.

Reviewer 2 Report

1.     The authors explored the direct and mediating effects of industrial restructuring and technological innovation on the evolution of carbon emission intensity. The theories, arguments, data, statistical regression, and analysis are reasonable. However, there are still some issues to be solved before this manuscript can be published.

2.     It seems that some of the English texts of this manuscript were directly translated from one or several Chinese reports, based on some terms and sentence structures. For example, line 45-47, line 61 (the industrial structure has become more industrialized…), and those (1), (2), (3)…in the texts such as the paragraphs lines 96-114 and lines 115-134.

3.     The last sentence in the first paragraph of the Literature Review seems problematic.

4.     It has been mentioned many times by the authors that carbon reduction is related to many factors, including industrial restructuring and others. However, a reduction in carbon emission is not equal to a reduction in carbon intensity. As the real indicator for carbon reduction is the percentage of carbon reduction derived from the absolute carbon emission instead of carbon intensity, I think the authors need to clarify these and avoid mixing the terms and concepts.

5.     For the theoretical analysis and research hypothesis, I suggest the authors can use some charts indicating the relationships between the key variables. These can help the readers understand the models and also the difference between the models.

6.     The variables shown in the equations and the texts need to be consistent in fonts and formats.

7.     The results and discussions from lines 325 to 540 are direct explanations of the models and results of the models. I suggest that some content related to research in other places/countries can also be referred to or compared.

8.     7.1 Conclusions and 7.2 Implications were presented using numbered items. I think these should be edited as essays instead of itemized texts. 

Author Response

Dear Editors and Reviewers:

Thank you very much for your comments about our paper submitted to International Journal of Environmental Research and Public Health (paper ID: ijerph-1921497).

We have learned much from your comments, which are fair, encouraging and constructive. After carefully studying the comments, we have made corresponding changes. Our response of the comments is enclosed at the end of this letter.

If you have any question about this paper, please don’t hesitate to contact us.

Sincerely yours.

The details of the revisions are as follows:

Point 1: It seems that some of the English texts of this manuscript were directly translated from one or several Chinese reports, based on some terms and sentence structures. For example, line 45-47, line 61 (the industrial structure has become more industrialized…), and those (1), (2), (3)…in the texts such as the paragraphs lines 96-114 and lines 115-134.

Response 1: Thank you very much for your suggestion, which has improved our article a lot. Thank you very much for your suggestions and for pointing out the problems. What you mentioned is not a direct translation from Chinese or related reports. It is a literature review we did, and the citation and narrative may be too simple, so we cut some unnecessary words and reorganized the narrative.

Point 2: The last sentence in the first paragraph of the Literature Review seems problematic.

Response 2: Thank you for your professional and constructive suggestion, it is very beneficial to enhance the credibility of our manuscript. We reworked the narrative in the last sentence of the first paragraph of the literature review. (Line 121-122, Page 3)

Point 3: It has been mentioned many times by the authors that carbon reduction is related to many factors, including industrial restructuring and others. However, a reduction in carbon emission is not equal to a reduction in carbon intensity. As the real indicator for carbon reduction is the percentage of carbon reduction derived from the absolute carbon emission instead of carbon intensity, I think the authors need to clarify these and avoid mixing the terms and concepts.

Response 3: This is a very helpful question for improving the logicality of our manuscript. Following your suggestion, we have scientifically explained in the article the definition of carbon intensity and why we chose to use it as a measure of carbon emissions. (Line 293-296, Page 7)

Point 4: For the theoretical analysis and research hypothesis, I suggest the authors can use some charts indicating the relationships between the key variables. These can help the readers understand the models and also the difference between the models.

Response 4: Thanks very much for your crucial comments. We have added a schematic diagram of the effects of industrial structural transformation, science and technology innovation on emission intensity to explain the relationship between the three more clearly. (Line 254-260, Page 6)

Point 5: The variables shown in the equations and the texts need to be consistent in fonts and formats.

Response 5: Thank you so much for your professional review work on our article and for giving some positive comments. We double-checked the errors you mentioned in the article regarding the font and formatting of the variables, and following your suggestions, we made the appropriate changes.

Point 6: The results and discussions from lines 325 to 540 are direct explanations of the models and results of the models. I suggest that some content related to research in other places/countries can also be referred to or compared.

Response 6: We feel great thanks for your review work on our article. We changed the conclusion section to Conclusion and Discussion, which explores how China compares to other countries in terms of relevant research. (Line 607-654, Page 16-17)

Point 7: 7.1 Conclusions and 7.2 Implications were presented using numbered items. I think these should be edited as essays instead of itemized texts.

Response 7: Thank you so much for your insightful and valuable suggestions. We revised part 7 of the article.

Reviewer 3 Report

Comments to the Author

The paper has an original idea. However, the way the article is written is hiding the added value of the work. The information provided is not adequate to justify publication. My comments are as follows: 

1. The authors have carried out a good research and added a significant knowledge to the world. But, title of the paper must be revised to make the paper more captivating to target audience. Also, abstract of the paper should be modified in order to clearly expound the novel contribution of authors. I first recommend to change the title of the paper to better reflect the causal nature and the context of the study. The term "Industrial Restructuring and Technological Innovation" is too vague and, if you want to keep it in the title, the list of the constructs should be complete.

 2. The objective of the research is grounded in a managerial problem created by "the mechanism of the mediating effect of technological innovation on industrial restructuring." with  by establishing fixed-effect model and a mediating effect model to empirically analyze the mechanism by which industrial restructuring and technological innovation affect carbon emission intensity. However, neither the theoretical background supporting the objective nor the research objectives are explained. The introduction contains a discussion of new formats and new models but I didn't understand the link with the preceding paragraphs or its relevance at this stage of the text. 

3. It is difficult to understand what academic contributions are made to the study. This paper explored the mechanism of the mediating effect of technological innovation on industrial restructuring and carbon reduction while accounting for the direct effect of industrial restructuring on carbon emissions. what is the methodological specificity and excellence used? Maybe meta-analysis can be performed in this research, when there are multiple scientific studies addressing the same question, with each individual study reporting measurements that are expected to have some degree of error and update of environmental policy through a weight and meta-analysis. The aim then is to use approaches from statistics to derive a pooled estimate closest to the unknown common truth based on how technological innovation is perceived. A meta-analysis is a statistical analysis that combines the results of multiple scientific studies. 

4. Is the paper's argument built on an appropriate base of theory, concepts, or other ideas? Has the research or previous literature on which the variables and models is based been well designed? Are the methods employed appropriate? 

5. The results are not well presented. The reader is confused when reading the results section. The discussion of the results is not provided. The link between the results and previous literature is missing. The regression results should be well articulated in the manuscript. An in-depth discussion should be given to support the purpose of the research. For all these reasons, I strongly suggest to major revise of this paper.

Author Response

Dear Editors and Reviewers:

Thank you very much for your comments about our paper submitted to International Journal of Environmental Research and Public Health (paper ID: ijerph-1921497).

We have learned much from your comments, which are fair, encouraging and constructive. After carefully studying the comments, we have made corresponding changes. Our response of the comments is enclosed at the end of this letter.

If you have any question about this paper, please don’t hesitate to contact us.

Sincerely yours.

The details of the revisions are as follows:

Point 1: . The authors have carried out a good research and added a significant knowledge to the world. But, title of the paper must be revised to make the paper more captivating to target audience. Also, abstract of the paper should be modified in order to clearly expound the novel contribution of authors. I first recommend to change the title of the paper to better reflect the causal nature and the context of the study. The term "Industrial Restructuring and Technological Innovation" is too vague and, if you want to keep it in the title, the list of the constructs should be complete.

Response 1: This is a very helpful question for improving the logicality of our manuscript. We changed the article title to “Study of the Impact of Industrial Restructuring on the Spatial and Temporal Evolution of Carbon Emission Intensity in Chi-nese Provinces--- Analysis of Mediating Effects Based on Technological Innovation”. At the same time, the abstract has been modified accordingly to provide the reader with all the innovative content and as much quantitative or qualitative information as possible in order to better present the results of the paper. (Line 2-38, Page 1)

Point 2: The objective of the research is grounded in a managerial problem created by "the mechanism of the mediating effect of technological innovation on industrial restructuring." with  by establishing fixed-effect model and a mediating effect model to empirically analyze the mechanism by which industrial restructuring and technological innovation affect carbon emission intensity. However, neither the theoretical background supporting the objective nor the research objectives are explained. The introduction contains a discussion of new formats and new models but I didn't understand the link with the preceding paragraphs or its relevance at this stage of the text.

Response 2: Thank you very much for your suggestion, which has improved our article a lot. Global warming has become one of the most hot issues of international concern in recent years, and the increasingly serious environmental problems have seriously restricted the high-quality development of human society. As an important economy, it is significant for China to realize low-carbon development. In the long run, industrial structure transformation can effectively reduce global energy consumption, and is also an important way to reduce carbon emission intensity. Therefore, the study of the impact of industrial structure transformation on carbon emission intensity has important theoretical value and practical significance. Based on this, we reorganized and supplemented the research objectives of this paper. (Line 93-104, Page 3)

Point 3: It is difficult to understand what academic contributions are made to the study. This paper explored the mechanism of the mediating effect of technological innovation on industrial restructuring and carbon reduction while accounting for the direct effect of industrial restructuring on carbon emissions. what is the methodological specificity and excellence used? Maybe meta-analysis can be performed in this research, when there are multiple scientific studies addressing the same question, with each individual study reporting measurements that are expected to have some degree of error and update of environmental policy through a weight and meta-analysis. The aim then is to use approaches from statistics to derive a pooled estimate closest to the unknown common truth based on how technological innovation is perceived. A meta-analysis is a statistical analysis that combines the results of multiple scientific studies.

Response 3: Thank you for your professional and constructive suggestion, it is very beneficial to enhance the credibility of our manuscript. We found that technological innovation and industrial structure transformation are two crucial factors affecting carbon emissions, but the existing studies have mainly examined the effects on carbon emissions from the perspectives of technological innovation and industrial structure transformation separately, without forming a unified framework, and this single-factor analysis model ignores the possible interaction between technological innovation and industrial structure upgrading on carbon emissions. The study area is more examined from the national level and less examined regional differences. Compared with the existing research literature, the main contribution of this study consists of two aspects: first, industrial structure transformation, science and technology innovation and carbon emission intensity are incorporated into a unified analytical framework, and the relationship between industrial structure transformation, science and technology innovation and carbon emission intensity is explained in two dimensions: industrial structure rationalization and industrial structure advanced. Second, with the help of Chinese provincial panel data, the possible interactions of the two on carbon emissions and regional variability are examined empirically using fixed-effects and mediating-effects models.

Point 4: Is the paper's argument built on an appropriate base of theory, concepts, or other ideas? Has the research or previous literature on which the variables and models is based been well designed? Are the methods employed appropriate?

Response 4: We deeply appreciate the reviewer’s suggestion. For the selection of arguments, variables and models, we study the relationship among industrial structure transformation, science and technology innovation and carbon emission intensity through the analysis of relevant domestic and foreign literature, following the principles of scientific and rationality, accessibility and comparability, and combination of qualitative and quantitative. In measuring the core indicators of industrial structure, science and technology innovation and carbon emission, we require both authentic and reliable data sources and the rationality of the selected indicators, so that the selected indicators scientifically and rationally reflect the variables of industrial structure, science and technology innovation and carbon emission intensity. For the selection of control variables, considering that carbon emission intensity will be affected by a variety of factors, the influence of each factor should be considered as comprehensively as possible when selecting the indicators. At the same time, it is essential to avoid choosing too numerous and complicated indicators, to select representative indicators, to use the minimum number of indicators to cover all dimensions, to convey as much information as possible, and to better reflect the influence of each factor on carbon dioxide emission intensity. At the same time, we also combine qualitative and quantitative analysis, select the corresponding indicators from a qualitative perspective, consider the influence of various factors comprehensively, and at the same time quantify the qualitative indicators and give clear definitions.

Point 5: The results are not well presented. The reader is confused when reading the results section. The discussion of the results is not provided. The link between the results and previous literature is missing. The regression results should be well articulated in the manuscript. An in-depth discussion should be given to support the purpose of the research. For all these reasons, I strongly suggest to major revise of this paper.

Response 5: Thanks for your constructive suggestion. We changed the conclusion section to conclusion and discussion, in which we compared the findings of the article with those of the existing literature and conclusions, explained the current findings of the article, and pointed out the shortcomings of this study as well as future research directions and ideas. (Line 607-670, Page 16-17)

Reviewer 4 Report

Thank you for this interesting  journal article. Can you please clarify few things before further investigation

What is the relation between this work with carbon neutralisation mentioned in the abstract?

How can you define the region of China?

Is there any further investigation for the economical characteristics for different region and province?

If so, can you please explain them in discussion and draw more finding by the model? 

Author Response

Dear Editors and Reviewers:

Thank you very much for your comments about our paper submitted to International Journal of Environmental Research and Public Health (paper ID: ijerph-1921497).

We have learned much from your comments, which are fair, encouraging and constructive. After carefully studying the comments, we have made corresponding changes. Our response of the comments is enclosed at the end of this letter.

If you have any question about this paper, please don’t hesitate to contact us.

Sincerely yours.

The details of the revisions are as follows:

Point 1: . What is the relation between this work with carbon neutralisation mentioned in the abstract?

Response 1: This is a very helpful question for improving the logicality of our manuscript. The Chinese government issued its "Opinions on the Complete and Accurate Implementation of the New Development Concept to Achieve Carbon Neutrality" in October 2021. As a significant and essential way to achieve carbon peaking and carbon neutrality, "deep adjustments of industrial structure" are considered necessary, and there are precise requirements for optimizing and upgrading industrial structures. The focus of industrial structure adjustment is to increase the proportion of tertiary industry and gradually reduce the proportion of secondary industry. On the other hand, achieving peak carbon and carbon neutrality is a complex project and a long-term task. Scientific and technological innovation will bring about changes in industrial structure, alter the operation of production factors, accelerate the transformation of scientific and technological achievements, and thus promote the transformation of economic growth patterns. Due to China's high-quality economic development, it is essential to study how upgrading industrial structures and technological innovations affect carbon emissions under environmental regulations. In order to achieve carbon neutrality and peak carbon, this is of enormous theoretical and practical importance.

Point 2: How can you define the region of China?

Response 2: We sincerely appreciate the reviewer for thoroughly reviewing our manuscript and providing helpful comments to guide our revision. Based on the division of China's three major regions by the China Bureau of Statistics (http://hprc.cssn.cn/wxzl/wxysl/wnjj/diqigewnjh/200907/t20090728_3954123.html), the sample is divided into eastern, central, and western regions. As a result of serious data deficiencies, Tibet was not included in the study. The eastern region includes 11 provincial-level administrative regions, namely Beijing, Tianjin, Hebei, Liaoning, Shanghai, Jiangsu, Zhejiang, Fujian, Shandong, Guangdong, and Hainan; the central region includes eight provincial-level administrative regions, namely Heilongjiang, Jilin, Shanxi, Anhui, Jiangxi, Henan, Hubei, and Hunan; the western region includes 11 provincial-level administrative regions, namely Sichuan, Chongqing, Guizhou, Yunnan, Shaanxi, Gansu Qinghai, Ningxia, Xinjiang, Guangxi, Inner Mongolia.

Point 3: Is there any further investigation for the economical characteristics for different region and province?  If so, can you please explain them in discussion and draw more finding by the model?

Response 3: Thank you for your professional and constructive suggestion, it is very beneficial to enhance the credibility of our manuscript. There are differences in socioeconomic development between the 30 provinces in the study area (autonomous regions and municipalities directly under the central government), such as resource endowments, geographical location, and industrial restructuring. Each province's industrial structure development and carbon emissions intensity levels are characterized by heterogeneity, both geographically and spatially. In order to more accurately investigate the impact of industrial structure transformation on regional carbon emission intensity levels, we divided the 30 research provinces into three regions: the eastern region, the central region and the western region. Using regional heterogeneity as a framework, we examined the impact of industrial structure transformation on regional carbon emission intensity.

Round 2

Reviewer 3 Report

Thanks for the correct revision, the quality of the paper is greatly improved enough to be published in the journal

Reviewer 4 Report

Thank you very much. 

Can you please seperate discussion and conclusion. Discussion is focusing on the implications of the results, and conclusion is the conclusion of the whole paper.

Also, can you upload a clean version and highlighed in yellow for what you have changed. It is very unclear now.